# Recent Findings Related to Cardiomyopathy and Genetics

**DOI:** 10.3390/ijms222212522

**Published:** 2021-11-20

**Authors:** Takanobu Yamada, Seitaro Nomura

**Affiliations:** Department of Cardiovascular Medicine, Graduate School of Medicine, The University of Tokyo, Tokyo 113-8654, Japan; taka0701.yamada@gmail.com

**Keywords:** cardiomyopathy, hypertrophic cardiomyopathy (HCM), dilated cardiomyopathy (DCM), genetics, genotype–phenotype correlations, pathogenesis

## Abstract

With the development and advancement of next-generation sequencing (NGS), genetic analysis is becoming more accessible. High-throughput genetic studies using NGS have contributed to unraveling the association between cardiomyopathy and genetic background, as is the case with many other diseases. Rare variants have been shown to play major roles in the pathogenesis of cardiomyopathy, which was empirically recognized as a monogenic disease, and it has been elucidated that the clinical course of cardiomyopathy varies depending on the causative genes. These findings were not limited to dilated and hypertrophic cardiomyopathy; similar trends were reported one after another for peripartum cardiomyopathy (PPCM), cancer therapy-related cardiac dysfunction (CTRCD), and alcoholic cardiomyopathy (ACM). In addition, as the association between clinical phenotypes and the causative genes becomes clearer, progress is being made in elucidating the mechanisms and developing novel therapeutic agents. Recently, it has been suggested that not only rare variants but also common variants contribute to the development of cardiomyopathy. Cardiomyopathy and genetics are approaching a new era, which is summarized here in this overview.

## 1. Introduction

Cardiomyopathy is a relatively rare and refractory myocardial disease caused by genetic and environmental factors. Cardiomyopathy is classified into dilated cardiomyopathy (DCM), hypertrophic cardiomyopathy (HCM), arrhythmogenic right ventricular cardiomyopathy (ARVC), restrictive cardiomyopathy (RCM), and unclassified cardiomyopathy [1]. HCM is the most common genetic heart diseases characterized by left ventricular hypertrophy that can lead to heart failure and sudden cardiac death. The prevalence of HCM in the general population has been estimated to range from 1:200 to 1:500 for asymptomatic cases and 1:3000 for symptomatic cases [2,3]. DCM is also an important genetic heart disease characterized by left ventricular dilation and systolic dysfunction. This disease is one of the leading causes of heart failure and requires cardiac transplantation in severe cases. The prevalence of DCM is estimated to be 1:2500 in the general population [4,5]. Since the discovery of pathogenic variants in the myosin heavy chain 7 (*MYH7*) gene in HCM [6] and in the cardiac alpha-actin (*ACTC1*) gene in dilated DCM [7], more than 100 causative genes have been reported to be causal for cardiomyopathy [8]. Clarification of the genotype–phenotype correlation is needed to develop the precision medicine in cardiomyopathy. In addition, elucidation of the mechanisms involved in cardiomyopathy onset and progression from the causative genes will lead to the development of novel therapeutic agents. In this review, we summarize the current genotype–phenotype correlations and the latest findings on pathogenesis and drug discovery for DCM and HCM. We also discuss the recent evidence regarding other cardiomyopathies such as peripartum cardiomyopathy, cancer therapy-related cardiac dysfunction, and alcoholic cardiomyopathy, and the significance of common variants in cardiomyopathy.

## 2. Association between Causative Genes and Phenotypic Features

Cardiomyopathy is a clinically heterogenous disease, and one of the factors that differentiates clinical phenotypes is genotype. The genotype–phenotype correlation has been studied for a long time, and the results of these studies are gradually being utilized in clinical practice. Here, we summarize the genotype–phenotype correlations in the representative cardiomyopathies, DCM and HCM.

### 2.1. Dilated Cardiomyopathy

Recent comprehensive targeted sequencing studies have identified pathogenic variants in approximately 40–50% of DCM patients [9,10]. The causative genes of DCM are diverse, and the representatives are shown in Table 1. The genotype–phenotype correlation in DCM has been widely studied. The major causative variants are titin (*TTN*)-truncating variants (TTNtv) and lamin A/C (*LMNA*) variants [9,10], and their clinical characteristics are gradually becoming clearer.

The clinical phenotype of patients with TTNtv is characterized by low penetrance, a low rate of conduction defects, relatively good clinical prognosis, and sex differences in prognosis (i.e., worse prognosis in males versus females) [9,10]. The odds ratio of developing DCM is higher in ethnic Europeans than in ethnic Africans [11], and TTNtv causes not only DCM, but also juvenile atrial fibrillation [12]. Moreover, patients with TTNtv tend to respond well to appropriate medical therapy, with a dramatic improvement in cardiac function, even if they had markedly reduced left ventricular function at diagnosis [10].

*LMNA* variants are the second most common cause of DCM [13] and are found in 5–10% of DCM patients [9,10]. The clinical phenotype of patients with *LMNA* variants is characterized by high penetrance, young age, conduction defects, poor prognosis, poor response to medical therapy, and high dependence on heart transplantation [9,10,14]. Given the high rate of conduction defects, the European Society of Cardiology guidelines recommend that DCM patients with *LMNA* pathogenic variants be treated with implantable cardioverter-defibrillators to prevent sudden death [15].

In a recent large cohort study [16], targeted next-generation sequencing of cardiomyopathy-related genes was performed in 1005 DCM patients, and the differences in the clinical course according to genotype were examined. Cardiovascular events were more frequently observed in the genotype-positive group, while left ventricular reverse remodeling (LVRR), in which the left ventricular ejection fraction improved, was less frequent in the genotype-positive group. Among the genotype-positive group, LVRR occurred more frequently in patients with *TTN* pathogenic variants, and cardiovascular events occurred more frequently in the nuclear envelope gene group including *LMNA*. This report also showed that the desmosomal gene group and the cytoskeleton/Z-disk gene group were at lower risk than the nuclear envelope group, but at higher risk than the other functional gene groups.

### 2.2. Hypertrophic Cardiomyopathy

Pathogenic variants are identified in about half of HCM patients [17,18,19], with most of the causative genes encoding sarcomere component proteins. The representatives are shown in Table 2.

In contrast to DCM, since the causative genes of HCM are functionally similar, the clinical course of HCM had been compared by dividing the patients into two groups: those with pathogenic variants in sarcomere gene and the others. Patients with pathogenic variants in sarcomere genes exhibit a younger age of onset and greater left ventricular wall thickness than those without such variants [20]. A recent registry study has shown that patients with pathogenic variants in sarcomere genes have a higher incidence of adverse events such as heart failure, ventricular arrhythmia, and atrial fibrillation, and that pathogenic variants in sarcomere genes are an important predictor of these adverse outcomes [21]. A recent report proposed a sudden cardiac death (SCD) risk prediction model for predicting SCD over five years in pediatric HCM and included the presence of pathogenic variants [22].

Sarcomere genes are classified into thick filament genes such as myosin-binding protein C3 (*MYBPC3*) and *MYH7* and thin filament genes such as troponin I (*TNNI3*) and troponin T (*TNNT2*). The clinical features related to these classifications have been investigated in recent years. In a study comparing the clinical course of two groups of HCM patients with thick and thin filament gene variants, patients with thick filament pathogenic variants tended to have greater hypertrophy and a larger left ventricular outflow tract pressure gradient, whereas those with thin filament pathogenic variants tended to have stronger diastolic dysfunction and a lower left ventricular ejection fraction (LVEF) [23]. In this study, the frequency of progression to heart failure in NYHA class III or IV was also reported to be higher in the thin filament variant positive group. In addition, research focusing on HCM with left ventricular systolic dysfunction (HCM-LVSD), in which the LVEF decreases to less than 50%, reported that the presence of pathogenic variants in the thin filament gene is a predictor of LVSD incidence among HCM patients [24]. 

Although there is no clear trend in the phenotypic characteristics of each causative gene, a meta-analysis determined that *MYH7* is frequently associated with ventricular arrhythmias and conduction defects [25].

## 3. Mechanisms of Cardiomyopathy Based on Causative Genes

Because it has become clear that cardiomyopathy is caused by genetic variants, it has become important to elucidate the pathogenesis of the cardiomyopathy for each variant. This has led to the development of analyses using patient-derived induced pluripotent stem cells (iPSCs), transgenic mice, and patient tissues. The *MYBPC3* gene is a major causative gene of HCM that encodes myocardial myosin-binding protein C. Analysis of truncating variants of this gene using iPSCs established from patients revealed that iPSC-derived cardiomyocytes (iPS-CMs) with mutations had abnormal calcium handling and enhanced nonsense-mediated mRNA decay [26]. Suppression of the *UPF1* gene, which regulates nonsense-mediated mRNA decay, rescued the calcium handling abnormalities, suggesting that activation of this pathway plays a key role in the pathogenesis. Furthermore, the function of isolated cardiomyocytes was analyzed using mice with truncating variants in the *MYBPC3* gene (MYBPC3tv) [27]. The results indicated that cardiomyocytes with MYBPC3tv had hypercontractility and inadequate relaxation and that treatment with the myocardial-specific ATPase inhibitor MYK-461 restored myocardial function [28]. In this context, structural changes in myosin, which regulates sarcomere contraction while consuming ATP, are restored after MYK-461 treatment [29]. This compound, named mavacamten, improved exercise tolerance, the post-exercise pressure gradient in the left ventricular outflow tract, NYHA class, and health status in a multicenter clinical study [30].

*TTN* is the major causative gene of DCM, and the titin encoded by this gene acts like a spring that ties myosin to the Z-band in sarcomeres. Pathogenic variants in this gene are concentrated in the A-band region [31], and DCM caused by truncating variants in this region has a relatively good prognosis [9,10]. iPSCs from patients with cardiomyopathy and TTNtv that were differentiated into cardiomyocytes exhibit impaired sarcomere formation and decreased myocardial contraction [32]. Live imaging analysis using patient-derived iPSCs revealed that the binding of the mutant titin protein to myocardial β-myosin (β-MHC) is the cause of the defective sarcomere formation [33]. Ribosome profiling showed that TTNtv induces nonsense-mediated mRNA decay in mutant alleles, leading to abnormalities in cardiac metabolism [34].

In addition, variants in *RBM20*, which encodes an RNA-binding splicing factor for several genes, including *TTN*, also cause DCM. There are two isoforms of the *TTN* transcript, N2B and N2BA. Compared with N2B, N2BA has a longer PEVK domain and a spring-like Ig domain, and functions as a long, soft spring. Pathogenic variants of *RBM20* are thought to decrease the N2B transcript of *TTN* and increase that of N2BA, resulting in the development of DCM by reducing active and passive tension in cardiomyocytes [35]. Recently, a new isoform of the *TTN* transcript, Cronos, which has a transcriptional start site slightly upstream of the A-band region of the *TTN* locus, was found to be expressed in cardiomyocytes during development and to promote sarcomere formation, suggesting that it is partly responsible for the defective sarcomere formation in TTNtv [36].

The *LMNA* gene encodes lamin A/C protein, which constitutes the nuclear lamina and is a causative gene of not only cardiomyopathy but also myopathy, lipoatrophy, and progeria. Analysis of patient-derived iPSCs and mutant model mice has shown that *LMNA* mutations induce activation of PDGF signaling [37] and DNA damage/p53 signaling [38] and modulate function of BRD4 (transcriptional regulator) [39], LSD1 (histone demethylase) [40], and EZH2 (polycomb complex component) [41], resulting in abnormalities in the epigenome and consequently changes in chromatin structure [42,43]. However, the mechanism of the DCM caused by *LMNA* mutations is not fully understood. Nuclear lamina also acts as a mechanosensor that transmits mechanosignals from outside the cell to the nucleus. Loss of function of the *LMNA* gene has been reported to lead to nuclear membrane disruption and the transfer of DNA repair molecules to the cytoplasm, resulting in DNA damage, telomere shortening, and cell cycle arrest [44]. In addition, close follow-up of the clinical profile of patients with *LMNA* mutations revealed an inverse correlation between the intensity of exercise habits and LVEF, suggesting that intense hemodynamic stress on the heart can induce cardiac dysfunction [45].

Recent studies have included not only functional analyses of each causative gene variant, but also cross-sectional analyses of several pathogenic variants. Bulk RNA-seq analysis of endomyocardial biopsy specimens from cardiomyopathy patients showed that gene expression greatly depended on the pathogenic variant involved, including TTNtv, the *LMNA* variant, the *RBM20* variant, and the *MYH7* variant [46]. In addition, integration of genetic variants and cardiac magnetic resonance imaging (MRI) revealed characteristic subepicardial ring-like scarring and regional contractile abnormalities in DCM caused by variants in the desmoplakin (*DSP*) and filamin C (*FLNC*) genes [47]. In addition to the *LMNA* gene, desmosome-related genes such as *PKP2*, *DSC2*, *DSP*, *DSG2*, and *JUP* were also identified as genetic variants associated with the appearance of the ventricular arrhythmias and sudden death observed in DCM [48].

As described above, the pathogenesis of each genetic variant is being elucidated, and therapeutic drugs to control the function of sarcomeres, which regulate the contractility of cardiomyocytes, are beginning to be developed. Although the objective of some drugs is to regulate sarcomeres through cyclic AMP signaling, they increase intracellular calcium levels, oxygen consumption, and heart rate, resulting in elevated mortality. However, the compound omecamtiv mecarbil (OM) can directly improve sarcomere function by strengthening myosin–actin binding and was expected to solve the abovementioned problems. Recently, the results of a phase III clinical trial examining the efficacy of this drug in chronic heart failure were reported [49]. A total of 8256 patients with symptomatic chronic heart failure with a LVEF less than 35% who were receiving standard heart failure therapy were randomly assigned to an OM group or placebo group, and adverse events occurring during a mean observation period of 21.8 months were analyzed. The primary endpoint was first heart failure event or cardiovascular death. During the observation period, the primary endpoint occurred in 1523 of 4120 patients (37.0%) in the OM group and in 1607 of 4112 patients (39.1%) in the placebo group (hazard ratio 0.92, 95% confidence interval 0.86–0.99, *p* = 0.03). There were no significant differences in cardiovascular death or Kansas City Cardiomyopathy Questionnaire (KCCQ) scores between the two groups. NT-proBNP was 10% lower in the OM group and cardiac troponin I was 4 ng/L higher in the OM group. It had been pointed out that this drug, which strengthens myosin–actin binding, might increase myocardial ischemia due to insufficient coronary blood flow, because diastolic time shortens with prolonged systolic ejection time. However, there was no significant difference in the occurrence of myocardial ischemia or ventricular arrhythmia between the two groups. Furthermore, considering the results of the subgroup analysis, which showed that the effect of OM was particularly large in the group with a LVEF less than 28% and high NT-proBNP, OM can be positioned as a drug that can be effective in patients with severe cardiomyopathy.

## 4. Other Cardiomyopathies and Genetics

In recent years, it has been reported that genetic background may contribute to the onset and progression of not only dilated and hypertrophic cardiomyopathy, but also other cardiomyopathies, which was previously thought to be caused mainly by environmental factors. In this review, we discuss three other cardiomyopathies that have been reported to correlate with pathogenic variants of DCM. 

### 4.1. Peripartum Cardiomyopathy

Peripartum cardiomyopathy (PPCM) is a cardiomyopathy with contractile dysfunction that develops in late pregnancy or the early postpartum period and occurs in less than 1:2000 deliveries [50,51]. The pathogenesis of PPCM remains unclear, but possibilities include fetal autoimmunity, microchimerism, myocarditis, excess dietary salt, or selenium deficiency [50,51,52]. In 2016, target sequencing of DCM-associated genes in 172 PPCM patients showed that TTNtv was significantly more common than in the general population and exhibited comparable prevalence to that in DCM [53]. In this study, patients with TTNtv had a lower ejection fraction at one year compared with those without TTNtv, suggesting that genetic variants in DCM-related genes may contribute to the development and progression of PPCM [53]. Larger-scale research in PPCM was subsequently performed [54]. The work involved target sequencing of cardiomyopathy-related genes in 469 PPCM patients and found that not only TTNtv, but also DCM-related genes such as *FLNC*, *DSP*, and *BAG3* were present at higher rates compared with the general population. PPCM patients with TTNtv had a significantly lower left ventricular ejection fraction at diagnosis than those without TTNtv, but there were no significant differences in the timing of onset, frequency of preeclampsia, or clinical improvement ratio [54]. The genetic background of PPCM may involve many factors, not only TTNtv, and is expected to be clarified by further large-scale and detailed research.

### 4.2. Cancer Therapy-Related Cardiac Dysfunction

Cancer therapy-related cardiac dysfunction (CTRCD), recognized as a decreased LVEF with or without overt signs or symptoms of heart failure [55], can develop during, immediately after, or many years after cancer treatment, affecting long-term prognosis [56,57,58]. This disease is becoming more and more important problem due to the improved treatment outcomes for malignancy. CTRCD is clinically heterogeneous, and the clinical course varies from transient myocardial damage to an irreversible one. The most common causative drugs that can cause irreversible myocardial damage are anthracyclines. Anthracycline-induced impairment of topoisomerase 2β has been suggested to cause reactive oxygen species production, mitochondrial dysfunction, and deoxyribonucleic acid double-stranded breaks [59]. However, since the sensitivity to anthracyclines varied considerably among patients, genetic background could be considered as one of the factors to explain this difference. The sequencing of cardiomyopathy-associated genes in 213 CTRCD patients revealed more pathogenic variants in these genes, mainly TTNtv, than in the healthy volunteer and reference populations. Moreover, CTRCD patients with TTNtv were more likely to have heart failure and atrial fibrillation and less likely to recover from myocardial damage [60]. In this research, it was also demonstrated that anthracyclines treatment in mice with heterozygous TTNtv resulted in lower left ventricular systolic dysfunction compared to wild type mice. These results suggest that patients with TTNtv may be more likely to develop CTRCD by cancer therapy and have a more severe cardiac phenotype. By identifying genetic risk factors, we will be able to identify cancer patients at high risk for CTRCD and provide preventive treatment. Further studies are needed to determine whether risk stratification by genetic testing can optimize cancer and cardiovascular therapies and reduce CTRCD while providing effective cancer treatment.

### 4.3. Alcoholic Cardiomyopathy

Alcoholic cardiomyopathy (ACM), which is caused by chronic and excessive alcohol intake, is characterized by left ventricular systolic dysfunction and dilation, as in DCM, but is distinguished by a relatively better prognosis than DCM. The left ventricular ejection fraction improved in about half of patients with ACM [61,62,63]. The involvement of DCM-related genes has also been reported in ACM [63]. Target sequencing of DCM-related genes in 141 ACM patients, 716 DCM patients, and 445 healthy controls revealed that ACM patients had more pathogenic variants in DCM-related genes such as TTNtv than healthy controls. The frequency of pathogenic variants in DCM-related genes was similar in ACM and DCM, and multivariate analysis showed that DCM patients with TTNtv and excessive alcohol intake had an 8.7% lower LVEF than those without TTNtv or excessive alcohol intake. However, ACMs with TTNtv did not differ from those without TTNtv in response to treatment for heart failure. As with the other cardiomyopathies, large-scale and detailed studies are needed to elucidate the association between genetic background and the development of ACM.

## 5. Multifactorial Aspects of Cardiomyopathy

Numerous studies have examined the relationship between cardiomyopathy and rare variants, and many associated rare genetic variants have been reported. However, in about half of these cardiomyopathy patients, the rare variant responsible for the disease remains to be identified. In addition, cardiomyopathy tends to have a lower penetrance than other monogenic diseases. To explain these problems, the role of common variants in cardiomyopathy has been reported, with the results of the first genome-wide association study (GWAS) of DCM in 2011 [64]. GWAS analysis of 517,382 SNPs in 1179 DCM patients and 1108 controls identified two DCM-associated SNPs, one of which was a non-synonymous variant in *BAG3*, which is a traditional DCM-related gene [64]. Three years later, GWAS analysis was performed on 4100 DCM patients and 7600 controls to identify new DCM-associated SNPs associated with the major histocompatibility complex [65]. Based on these results, a more extensive study was reported in 2020 [66]. GWAS analysis was performed to identify SNPs associated with cardiac function in 36,041 patients with available cardiac MRI findings who did not have a diagnosis of cardiac diseases from the UK Biobank. In total, 57 genetic loci associated with cardiac function were detected and the polygenic risk score using 28 SNPs associated with the left ventricular end-systolic volume index (LVESVi) was shown to be strongly correlated with the development of DCM [66]. GWAS analysis of HCM was also reported in 2021 [67]. GWAS analysis of 2780 HCM patients and 47,486 controls identified 12 HCM-associated SNPs, with a particularly strong polygenic influence in sarcomere-negative HCM [67]. Furthermore, another study of 1733 HCM cases and 6628 controls identified new HCM-associated loci near *HSPB7* and *BAG3*, which have been reported as risk alleles for DCM in previous studies and found to be protective in HCM [68]. Based on these results, the authors evaluated the results of GWAS analysis of nine left ventricular traits extracted from cardiac MRI of HCM patients, DCM patients, and the general population using LD score regression. The contractility and morphology parameters in the general population were correlated with both DCM and HCM patients, with opposing effects in the two diseases [68].

In conclusion, a polygenic background has been reported to be involved in the onset and progression of cardiomyopathy, which was once considered to be a monogenic disease caused by rare variants. Another GWAS analysis was conducted for entire heart failure, where environmental factors must have been largely involved. Twelve variants at 11 genomic loci, including a variant reported in DCM, were reported to be associated with heart failure [69]. The common variants, as well as rare variants, may be associated not only with the development of cardiomyopathy but also with its clinical course. Few GWAS and whole-genome sequencing analyses in cardiomyopathy have been reported, and it is expected that such studies will clarify in greater detail the genetic background of cardiomyopathy.

## 6. Clinical Indications for Genetic Testing

As we discuss, the relationship between genetic background and clinical phenotype in cardiomyopathy is rapidly being elucidated, but there is insufficient discussion on how to apply such information clinically. In general, implantable cardioverter defibrillators are recommended for DCM patients with *LMNA* pathogenic variants because of the high incidence of lethal arrhythmias. Such evidence may be revealed one after another, and it is important to perform genetic testing in order to practice precision medicine that relies on the results of genetic testing. Based on the above, a scientific statement summarizing the current best practice of genetic testing in cardiovascular medicine was published in 2020 [70]. According to this statement, genetic testing should usually be performed in patients with a confirmed or suspected diagnosis of inherited cardiovascular disease, and the family members with the most reliable and severe phenotype should be tested first. Following this approach, it is important to perform genetic testing on patients and to use its results for clinical management.

Genetic testing of asymptomatic relatives also needs to be considered. The statement in genetic testing states that genetic testing should also be performed on patients who are at high risk due to pathogenic variants already identified in their family members [70]. In a retrospective analysis of 285 individuals who had pathogenic variants of the sarcomere protein genes, known to be the causative genes of HCM, but did not meet the diagnostic criteria for HCM, approximately 50% were reported to develop HCM after 15 years of follow-up [71]. In addition, according to a large-scale study of 200,584 individuals in the general population, the prevalence of pathogenic variants of the sarcomere protein genes was 0.25%, and the risk of death and major adverse cardiac events was higher than in the control group [72]. It was also reported that individuals with a pathogenic variant of *LMNA*, a typical causative gene of DCM, had a cardiac penetrance close to 100% at age 60 [14]. These reports suggest that it is necessary to identify individuals with pathogenic variants of cardiomyopathy and to provide appropriate follow-up testing even if they are asymptomatic. Moreover, a study examining the cost-effectiveness of genetic testing in asymptomatic relatives of patients with DCM reported a 90% probability that the use of genetic testing for clinical surveillance of asymptomatic relatives would be cost-effective [73]. Further cost reduction of genetic testing is expected in the future, and genetic testing will become more important as new evidence on precision medicine in at-risk individuals becomes available.

## 7. Conclusions

Research related to cardiomyopathy and genetics have made remarkable progress, and high-throughput studies using NGS have successively elucidated these relationships. Genetic analysis may be not only useful for precision medicine in dilated and hypertrophic cardiomyopathy, but also important for preventing the onset and progression of other cardiomyopathies, such as PPCM and CTRCD, and in healthy individuals with pathogenic variants of cardiomyopathy-related genes. Elucidation of the new genetic factors may also allow for more accurate prediction of phenotypes and clinical courses. Through such studies, it is anticipated that the mechanisms underlying the onset and progression of cardiomyopathy and heart failure will be elucidated, and that new treatments will be developed.

## Figures and Tables

**Table 1 ijms-22-12522-t001:** Representative causative genes of dilated cardiomyopathy.

Functional Group	Gene	Protein
Sarcomere	*TTN*	Titin
	*ACTC1*	α-Cardiac actin
	*MYH7*	β-Myosin heavy chain
	*TNNC1*	Cardiac troponin C
	*TNNT2*	Cardiac troponin T
	*TNNI3*	Cardiac troponin I
Nuclear Envelope	*LMNA*	Lamin A/C
	*EMD*	Emerin
Desmosomes	*DSP*	Desmoplakin
	*DSG2*	Desmoglein 2
	*DSC2*	Desmocollin 2
Cytoskeleton	*DES*	Desmin
	*VCL*	Vinculin
	*FLNC*	Filamin C
Z-disk	*BAG3*	BCL2-associated athanogene 3
	*CSRP3*	Muscle LIM protein
	*MYPN*	Myopalladin
Ion channels	*SCN5A*	Sodium channel protein type 5
Sarcoplasmic reticulum	*PLN*	Phospholamban
	*RYR2*	Ryanodine receptor 2
Others	*RBM20*	RNA-binding protein 20

**Table 2 ijms-22-12522-t002:** Representative causative genes of hypertrophic cardiomyopathy.

Functional Group	Gene	Protein
Sarcomere (Thick Filament)	*MYBPC3*	Myosin-binding protein C
	*MYH7*	β-Myosin heavy chain
	*MYL2*	Myosin light chain 2
	*MYL3*	Myosin light chain 3
	*TTN*	Titin
Sarcomere (Thin Filament)	*TNNC1*	Cardiac troponin C
	*TNNT2*	Cardiac troponin T
	*TNNI3*	Cardiac troponin I
	*TPM1*	Tropomyosin α-1
	*ACTC1*	α-Cardiac actin
Others	*ACTN2*	α-Actinin 2
	*CSRP3*	Muscle LIM protein
	*FLNC*	Filamin C
	*PLN*	Phospholamban
	*ALPK3*	α-Protein kinase 3

## Data Availability

Not applicable.

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
