# Peer review of "Recent Findings Related to Cardiomyopathy and Genetics"

_ijms, 2021, doi:10.3390/ijms222212522_

Round 1
Reviewer 1 Report
The authors did an excellent job updating the field of cardiomyopathy genetics. The article describes genetics associated with primary and secondary cardiomyopathies, which is a strength as well as the recent discoveries of how common genetic variants may contribute.
Reviewer 2 Report
I read the manuscript. It seems remarkably clear of typographical or grammatical errors. The content is a satisfactory review of the genetic basis of cardiomyopathy. I would have liked to see some more information on the non Titin-related genetic forms associated with cardiomyopathy, in particular those associated with arrhythmogenic right ventricular cardiomyopathy which is mentioned but not discussed in detail.
Overall, this is a good review of a complex subject. It is difficult to cover the entire gamut of research on cardiomyopathies but this is a great summary.